# Interaction of Shock Waves with Water Saturated by Nonreacting or Reacting Gas Bubbles

**DOI:** 10.3390/mi13091553

**Published:** 2022-09-19

**Authors:** Sergey M. Frolov, Konstantin A. Avdeev, Viktor S. Aksenov, Illias A. Sadykov, Igor O. Shamshin, Fedor S. Frolov

**Affiliations:** 1Department of Combustion and Explosion, Semenov Federal Research Center for Chemical Physics of the Russian Academy of Sciences, 119991 Moscow, Russia; 2Institute of Laser and Plasma Technologies, National Research Nuclear University “Moscow Engineering Physics Institute”, 115409 Moscow, Russia; 3Department of Computational Mathematics, Federal State Institution “Scientific Research Institute for System Analysis of the Russian Academy of Sciences”, 117218 Moscow, Russia

**Keywords:** bubbly water, bubbly detonation, shock wave-to-bubbly water momentum transfer, bubble explosion, underwater propulsion

## Abstract

A compressible medium represented by pure water saturated by small nonreactive or reactive gas bubbles can be used for generating a propulsive force in large-, medium-, and small-scale thrusters referred to as a pulsed detonation hydroramjet (PDH), which is a novel device for underwater propulsion. The PDH thrust is produced due to the acceleration of bubbly water (BW) in a water guide by periodic shock waves (SWs) and product gas jets generated by pulsed detonations of a fuel–oxidizer mixture. Theoretically, the PDH thrust is proportional to the operation frequency, which depends on both the SW velocity in BW and pulsed detonation frequency. The studies reported in this manuscript were aimed at exploring two possible directions of the improvement of thruster performances, namely, (1) the replacement of chemically nonreacting gas bubbles by chemically reactive ones, and (2) the increase in the pulsed detonation frequency from tens of hertz to some kilohertz. To better understand the SW-to-BW momentum transfer, the interaction of a single SW and a high-frequency (≈7 kHz) sequence of three SWs with chemically inert or active BW containing bubbles of air or stoichiometric acetylene–oxygen mixture was studied experimentally. Single SWs and SW packages were generated by burning or detonating a gaseous stoichiometric acetylene–oxygen or propane–oxygen mixture and transmitting the arising SWs to BW. The initial volume fraction of gas in BW was varied from 2% to 16% with gas bubbles 1.5–4 mm in diameter. The propagation velocity of SWs in BW ranged from 40 to 580 m/s. In experiments with single SWs in chemically active BW, a detonation-like mode of reaction front propagation (“bubbly quasidetonation”) was realized. This mode consisted of a SW followed by the front of bubble explosions and was characterized by a considerably higher propagation velocity as compared to the chemically inert BW. The latter could allow increasing the PDH operation frequency and thrust. Experiments with high-frequency SW packages showed that on the one hand, the individual SWs quickly merged, feeding each other and increasing the BW velocity, but on the other hand, the initial gas content for each successive SW decreased and, accordingly, the SW-to-BW momentum transfer worsened. Estimates showed that for a small-scale water guide 0.5 m long, the optimal pulsed detonation frequency was about 50–60 Hz.

## 1. Introduction

A novel type of underwater propulsion device for producing hydrojet thrust, referred to as the pulsed detonation hydroramjet (PDH), was proposed in [1]. This propulsion device is a flow-through pulsed detonation tube inserted in a flow-through water guide. The pulsed detonation tube periodically (with a frequency of tens of hertz) generates shock waves (SWs) and produces gas jets due to ignition and deflagration-to-detonation transition in the combustible mixture filling the tube, and it transmits these SWs and product gas jets to water in the water guide, thus creating the compressible flow of bubbly water (BW) in it. The water guide takes pure outboard water through an intake and ejects BW through a nozzle. Hydrojet thrust is produced due to the acceleration of BW in the water guide by periodic SWs and product gas jets. Theoretically, the PDH thrust is proportional to the operation frequency depending on both the SW velocity in BW and on the frequency of pulsed detonations. The PDH can contain no moving parts and can be used for generating a propulsive force in large-, medium-, and small-scale underwater thrusters. Similar principles can potentially be used in microrobotics to provide the driving force [2] and in medicine for needle-free injections [3].

There are many publications on the processes inherent in PDH operation, including SW propagation in BW, submerged gas jet penetration and bubble formation, SW coalescence, etc. The basic phenomenon in the PDH is SW propagation in compressible BW. This phenomenon was earlier studied experimentally in a vertical hydroshock tube [4,5] containing a high-pressure chamber (HPC) and low-pressure chamber (LPC) separated by a bursting diaphragm, and a measuring section filled with BW with a certain volumetric gas content α. In experiments, after the rupture of the bursting diaphragm, an SW of known intensity (very weak, weak, or strong depending on the gas pressure in the HPC) was formed in the LPC and transmitted to BW. The evolution of SW velocity and other parameters in BW was monitored with pressure sensors mounted along the measuring section and with a high-speed video camera through optical windows. Experiments on very weak (quasiacoustic) SW propagation in BW with different α [6] showed that the speed of sound passed through a deep minimum as α increased from zero to 100% and was much less than the speed of sound in pure water (1500 m/s) and in pure air (340 m/s). This result corresponded well to the known theoretical relationships [7,8]. Experiments on weak SW propagation in water with air bubbles at α ranging from 1 to 20% showed [9] that the SW velocity in BW was supersonic and varied from 150 to 100 m/s when α increased from 2 to 5%, and it dropped to 50 m/s when α increased to 20%. The measured velocity of an SW reflected from a rigid wall appeared to be higher than that of the incident wave because of bubble shrinking in the BW compressed by the incident SW and therefore the lowering of α. Video recording of single bubble motion behind a weak SW in BW with α ≈ 1–3% showed [10] that the shock-induced bubble velocity attained a value of 3–4 m/s. Experiments with strong SWs in BW with α ranging from 0.5 to 6% showed [11] that the measured SW velocities were considerably greater than those registered in [9], other conditions being equal, and it attained 400 m/s at α = 2% and 250 m/s at α = 5%. Depending on BW parameters such as bubble size, gas thermal conductivity, liquid viscosity, etc., the SWs propagating in BW were found to exhibit various pressure profiles [12,13], namely, with a smooth or oscillatory pressure time history [14]. Experiments with SWs propagating in BW with α = 8% and air bubbles 0.1, 0.48, and 0.69 mm in diameter indicated [15] that the pressure oscillation frequency behind SWs decreased with bubble diameter. The authors of [16] used high-speed photography and image post-processing to register time-resolved structural changes in a submerged gaseous jet emanating from a Laval nozzle. In [17], the results of experimental study on gas jetting by an underwater detonation tube were reported, and the mechanism of shock wave propagation and bubble deformation was discussed. The effect of the nozzle attached to a detonation tube on the underwater SW and gas detonation bubble was investigated in [18]. Three types of nozzles (converging, straight, and diverging) were examined. The converging nozzle was shown to enhance water–gas mixing and increase the peak pressure of the SW compared with the straight nozzle as well as to essentially inhibit the bubble pulsation process. The diverging nozzle was shown to suppress water–gas mixing, increase the gas jet velocity, and enhance the bubble pulsation process. High-speed photography, digital particle image velocimetry, underwater pressure field measurements, and CFD calculations were used in [19,20] to study the two-phase flow nearby the open end of the detonation tube submerged in water. Stoichiometric explosive mixtures of methane, hydrogen, and acetylene with oxygen were detonated in the tube under the same fill conditions. The oscillation frequencies and directional growth of the detonation gas bubble were investigated. The dynamic behavior of the bubble in the first oscillation was found to be very similar to that of a conventional underwater explosion.

Theoretical and computational studies of SW propagation in BW were commonly based on one-dimensional conservation equations of mass, momentum, and energy for two mutually penetrating continua, water and gas. The simplest isothermal models of SWs in BW [15,21] assumed a small gas volume fraction in incompressible water and noninteracting spherical gas bubbles moving in water without velocity slip. The range of validity of the governing equations used in [15,21] was studied in [22]. Shock wave propagation in BW with both compressible phases was considered in [23]. A more detailed model of SW propagation in bubbly liquid accounting for the thermal conductivity of phases was proposed in [24]. The authors of [25] generalized the nonisothermal model formulation to an arbitrary number of fluids with interphase mass and energy transfer. The surface tension force at a curved interface of phases in BW was introduced in [26,27] to account for different pressures in phases. Shock-induced chemical energy release in SWs propagating through BW saturated with reactive gas bubbles was investigated numerically in [28,29]. The effect of bubble polydispersity on shock wave propagation in a bubbly liquid was investigated numerically in [30]. The averaged shock structure in one-dimensional calculations was shown to become less oscillatory and tending to monotonic when the bubble size distribution broadened. The authors of [31] computationally studied the interaction of shock waves generated by two underwater detonation tubes. The dynamics of detonation gas bubbles and spectral characteristics of pressure field were analyzed, and the formation of a high-pressure zone in the region between the tubes was revealed. In the up-to-date models, the governing equations are supplemented with the various semiempirical relationships for interphase mass, momentum, and energy exchange caused by shock-induced phase velocity slip and temperature differences.

Momentum transfer from a single strong SW to BW with chemically inert air bubbles was studied computationally [32,33] and experimentally [34,35]. It turned out that the highest efficiency of momentum transfer was achieved at a volumetric gas content α of about 20–25%. With such a gas content, the increment of the absolute velocity of BW behind a strong traveling SW attained a value as high as 30 m/s. The first low-frequency valveless and valved models of PDH were designed, manufactured, and tested in [36,37]. The performance of PDH models operating at a pulse generation frequency of up to 20 Hz was investigated in [38,39]. The time-averaged specific impulse of PDH models attained 550 s [40], which was higher than that of the most advanced liquid propellant rocket engines.

The possible directions of the improvement of PDH thrust performances are to use BW with bubbles of chemically active rather than inert gas (i.e., to increase the SW velocity in BW) and to increase the frequency of pulsed detonations from tens of hertz to some kilohertz (i.e., to replace pulsed detonations by continuously rotating detonations [41]). The former direction is substantiated by the following findings. When an SW penetrates a liquid containing bubbles of a chemically reactive gas uniformly distributed over the volume, “bubbly detonation”, that is a self-sustaining detonation-like solitary pressure wave propagating quasi-steadily at a supersonic velocity, may occur. Bubbly detonation was apparently observed for the first time in experiments [42] conducted in a hydroshock tube of a square 50 × 50 mm cross-section 1985 mm long and aimed at studying the interaction of SWs with a chain of reactive gas bubbles (a mixture of 70% Ar + 30% (2H_2_ + O_2_)) placed in glycerin. The bubble chain length was ≈670 mm, and the mean bubble diameter was ≈10 mm. Later, systematic experimental studies of bubbly detonation in water with bubbles of a reactive gas (stoichiometric acetylene–oxygen mixture) were conducted in a vertical hydroshock tube with an inner diameter of 35 mm and a total height of 5635 mm (the height of the BW column was 4195 mm) [43]. In experiments [44,45], bubbly detonation was initiated by transmitting gaseous detonation into the BW column with α of up to 10%. Bubbly detonation occurred only at α less than 6% (upper limit) at a considerable distance from the gas–BW interface attaining 2.5 to 3.5 m. At larger gas contents, bubbly detonation failed to occur. At very low gas contents (less than 0.5%), bubbly detonation was also not observed (lower limit). The influence of liquid viscosity on the limits of the existence of bubbly detonation in terms of α was studied experimentally in [46]. Experiments were conducted under conditions similar to those in [44,45]. An increase in liquid viscosity (by adding up to 50%vol glycerol to water) at α ranging from 1 to 6% made it possible to initiate bubbly detonation at a lower SW intensity (less than 1.7 MPa instead of 4–5 MPa). The main specific features of bubbly detonation were outlined in these studies. Firstly, its propagation velocity was always higher than the SW propagation velocity in a liquid with bubbles of a chemically inert gas under similar conditions and higher than the speed of sound in bubbly liquid. For example, in [44] at α ≈ 2%, the propagation velocities of bubbly detonation and SW, as well as the speed of sound in BW were ≈560 m/s, ≈425 m/s, and ≈85 m/s, respectively. Secondly, the process of propagation of bubbly detonation was self-sustaining [44], whereas SWs in a liquid with bubbles of a chemically inert gas gradually decayed. For using bubbly detonations in the PDH (e.g., for boosting thrust), it is necessary to know whether it is possible to obtain bubbly detonation in water without thickening additives at short distances (less than about 1 m) in a wide range of α and whether bubbly detonation gives any noticeable effect as compared to a SW. As for the increase in the frequency of pulsed detonations from tens of hertz to some kilohertz, one has to take into account that the volumetric gas content and mean bubble diameter in BW filling the PDH water guide may depend on the operation frequency, so that the initial parameters of BW in each subsequent operation cycle will be determined by the parameters of previous cycles. Therefore, there is a need in detailed studies to measure the effect of the SW generation frequency on the flow pattern in the PDH water guide and on the efficiency of SW-to-BW momentum transfer. No such studies are available in the literature.

In view of these two possible directions of the improvement of PDH thrust performances, the objective of the present work was twofold. On the one hand, the phenomenology of single SW propagation in pure water saturated with nonreacting or reacting bubbles, specific features of SW-to-BW momentum transfer, and the phenomenology of propagating bubbly detonations was studied experimentally. On the other hand, the specific features of the interaction of high-frequency (≈7 kHz) SW pulses with water saturated with air bubbles was studied experimentally. These objectives, as well as the obtained results, are the novel and distinctive features of the present work.

## 2. Materials and Methods

### 2.1. Test Rig for Studies of Single Shock Wave Propagation in Bubbly Water

Figure 1a shows a schematic of the test rig used in experimental studies of single SW propagation in BW. A vertical hydroshock tube with a cross-section of 50 × 100 mm consisted of the HPC and LPC separated by a bursting diaphragm, two optical sections with Plexiglas windows (a slit 10 × 200 mm in the upper section of LPC and six rectangular windows 55 × 55 mm in the lower section of LPC), and a bubble generator. The overpressure in HPC is measured by the low-frequency pressure sensor P0 (Metronic KURANT-DI200). The error of determining the overpressure in HPC was estimated at 1%. Along the LPC, 6 piezoceramic pressure sensors P1 to P6 were installed (four KISTLER 211B2 and two PCB 113B24) flush with the inner surface of the tube. The natural frequency of pressure sensors P1 to P6 was 500 kHz. The error of determining the pressure amplitude in experimental conditions was estimated at 10%. To maintain atmospheric pressure, *P*_0,LPC_ = 0.1 MPa, above the BW column surface, the LPC had a hole 3 mm in diameter at a distance of 1 cm below the diaphragm. To obtain contrast video frames, video recording was carried out in backlight.

The bubble generator was made in the form of a steel plate 9 mm thick, in which 50 capillaries with an inner diameter of 0.26 mm were mounted in the nodes of a square grid with a step of 10 mm. Gas (air or a premixed C_2_H_2_ + 2.5 O_2_) was supplied to the bubble generator from the receiver with a volume of 6 L at a given pressure through the solenoid valve. The volumetric content of gas, α, in BW depended on the initial pressure in the receiver, the value of which was determined during preliminary experiments. For the sake of convenience, in what follows, water with air bubbles will be referred to as inert BW, and water with bubbles of a reactive gas mixture will be referred to as active BW.

The experimental procedure was as follows. Based on the required α, the height of the water column, h − Δh = h(1 − α), was calculated. Thereafter, the LPC was filled with water up to the required level. Next, a bursting diaphragm consisting of several sheets of a 50-micron polyethylene terephthalate film was installed between the HPC and LPC. The number of film sheets in experiments varied from 2 to 10 depending on the target pressure in the HPC, P0, HPC. After evacuation, the HPC was filled with a combustible mixture (C_2_H_2_ + 2.5 O_2_) up to pressure P0,HPC. Then, the control and data acquisition systems were put into a standby mode, and a signal was sent from the remote control to the control system, which first opened the solenoid valve for gas supply from the receiver to the bubble generator for a time sufficient to form uniform BW over the entire water column height (≈8 s in the present experiments), and then, it simultaneously issued a synchronization signal to ignition, video camera (Phantom Miro LC310), and analog-to-digital converter (ADC, R-Technology QMBox QMS20).

### 2.2. Test Rig for Studies of Shock Wave Package Propagation in Bubbly Water

Figure 1b shows a schematic of the test rig used in experimental studies of SW package propagation in inert BW. In general, this test rig was somewhat similar to that shown in Figure 1a but differed from it in that the HPC was replaced by the SW generator and the LPC was equipped with more windows for optical access. The SW generator was installed in the upper part of the measuring section and was used for generating a series of three successive SWs in inert BW. For the sake of clarity, Figure 2 shows the photograph with the explosive view of the SW generator. The SW generator consisted of a curved donor detonation tube 20 mm in inner diameter and 1250 mm long and three acceptor detonation tubes attached to it, which were each 0.9 m long. The donor detonation tube included a predetonator installed at the tube inlet, a KISTLER 211B2 piezoceramic overpressure sensor installed at a distance of 200 mm from the tube inlet, and a bursting diaphragm installed at the tube outlet for preventing water suction into the tube after the expansion of detonation products to BW.

The experimental procedure was as follows. First, the required value of α was adjusted as described in Section 2.1. Then, the donor and acceptor detonation tubes were purged with air for 1 min with the removed bursting diaphragm to relief detonation products remaining from the previous experiment and filled with the stoichiometric propane–oxygen mixture. Next, the bursting diaphragm was installed, and the tube was blown with a combustible mixture until a control signal was sent by the control system to simultaneously trigger the ignition system, the data acquisition system (based on the QMBox QMS20 ADC), and the high-speed video camera. After ignition, a detonation wave was initiated in the predetonator, which then propagated along the donor detonation tube and, passing through the attachment points of acceptor detonation tubes 1–3 (see insert in Figure 1b), branched into a detonation wave running along the donor detonation tube and a detonation wave running along the corresponding acceptor detonation tube. Taking into account that the tube branching did not actually affect the detonation velocity *D* in its different parts, detonation in the second acceptor detonation tube occurred with a delay τ = *L*_d_/*D* with respect to the first acceptor detonation tube, whereas in the third acceptor detonation tube, it occurred with the same delay with respect to the second acceptor detonation tube (here, *L*_d_ is the distance between the corresponding cross-sections of the donor detonation tube branching), providing the same delay in the release of detonation from the acceptor tubes into BW. The experimental value of detonation velocity in the acceptor detonation tubes was obtained by processing the pressure records of sensor P2 and video records (see below).

## 3. Results and Discussion

### 3.1. Single Shock Wave Propagation in Bubbly Water

To obtain the gas pressure in the HPC at a level feasible for hydrojet propulsion (up to 6–8 MPa [1]), the HPC was filled by the stoichiometric acetylene–oxygen mixture, which was ignited and burned to rupture the bursting diaphragm. The initial pressure of the mixture in the HPC was *P*_0,HPC_ = 0.4–0.6 MPa. In all experiments, prior to diaphragm rupture, the LPC was filled with an inert or active BW to a height of ≈196 cm with bubbles of initial diameter *d*_0_ = 1.5–4 mm. The remaining part of the LPC with a height of 2–5 cm above the BW column surface was filled with gas at atmospheric pressure. The initial volume fraction of gas in BW, α, varied from 2.0 ± 0.1% to 10.0 ± 0.5%. The air and water were at room temperature *T*_0_ = 298 K.

Figure 3 and Figure 4 show the records of pressure sensors P1 to P6 in two experiments conducted under the same initial conditions with inert and active BW, respectively. The time was counted from the launch of synchronization signal. At the initial stage of SW propagation (see records of sensor P1 in the bottom of Figure 3 and Figure 4), the amplitude and shape of the curves in both cases were very similar: the average SW amplitude was ≈7 MPa, which was superimposed by pressure fluctuations with a frequency of 23–27 kHz. In the SW propagating through inert BW, the amplitude of pressure fluctuations reached 15 MPa at sensor P2 and decreased with time attaining ≈5 MPa at sensor P6, while the characteristic frequency of pressure fluctuations was 25–50 kHz. This frequency was close to the frequency of transverse acoustic oscillations of the BW column. The SW front in inert BW on all sensors was gentle with a duration of 0.2–0.3 ms. Pressure fluctuations in the records of Figure 3 were mainly observed within ≈200 µs after the passage of the SW front. When the SW propagated through active BW (see records of sensors P2 to P6 in Figure 4), the peak pressure fluctuations in the wave were significantly higher (up to 22 MPa) than in Figure 3 and did not decrease with time. The characteristic frequency of pressure fluctuations was very high (100–500 kHz) and approached the natural frequency of pressure sensors. As in Figure 3, pressure fluctuations in the records of Figure 4 were mainly observed within ≈200 µs after the passage of the SW front; however, the duration of the peak intensity of pressure pulsations was only 20–40 µs. The pressure wave front in this case had a much shorter duration (0.02–0.1 ms vs. 0.2–0.3 ms) than in Figure 3.

Figure 5 shows a sequence of video frames illustrating SW propagation through active BW at α = 2%. Video recording was made through the top three windows of the lower optical section of the LPC (see Figure 1a). Shock wave propagation in active BW is accompanied with shock-induced explosions of individual bubbles, which appear as multiple luminous spots behind the propagating SW. The apparent propagation velocity of the luminocity front determined by its slope to the horizontal line is seen to be nearly constant. When such a SW–luminosity front complex reflects from the bottom of the hydroshock tube, the arising pressure is so high that the 9 mm thick steel plate of bubble generator is bent down by 3 mm (see insert in Figure 1a).

Figure 6 compares the measured dependences of the average SW velocity on the distance traveled in inert and active BW at two values of α: 2% (Figure 6a) and 10% (Figure 6b). Although the SW structure in active BW did not contain a solitary wave inherent in bubbly detonation [42,43,44,45,46] (see Figure 4), the SW velocity in active BW was approximately 100 m/s (at α = 2%) and 50 m/s (at α = 10%) higher than in inert BW with other conditions being equal. In other words, the replacement of inert bubbles with active bubbles increased the SW velocity significantly: by 20–30%. The increase in the SW propagation velocity in active BW was obviously caused by chemical energy deposition due to shock-induced explosions of individual bubbles. At a segment between 600 and 1200 mm, the SW velocity in active BW was about constant and equal to ≈500 m/s at α = 2% and ≈270 m/s at α = 10% (horizontal dashed lines in Figure 6a,b, respectively). A further decrease in the SW velocity with distance could be attributed to the nonuniform structure of BW closer to the bubble generator near the tube bottom, in particular at α = 2%. As compared to the bubbly detonation velocity measured in [44] for active BW with bubbles of acetylene–oxygen mixture at α = 2% (≈650 m/s), a constant propagation velocity of 500 m/s in the present experiments was 23% less, but it was attained at a considerably shorter distance: 600 mm vs. 2300 mm in [44]. The latter is important for practical applications in the small-scale PDH, because the elevated SW velocity could allow increasing the PDH operation frequency and thrust. As the constant-velocity mode of SW propagation in active BW is associated with shock-induced explosions of individual bubbles, this mode will be further referred to as bubbly quasidetonation.

Figure 7 is plotted for better understanding the difference between SW propagation velocities in inert and active BW. It compares video frames of experiments with inert and active BW at α = 2%. As in Figure 5, video recording was made through the top three windows of the lower optical section of the LPC. Note that pressure sensor P5 was installed 65 mm above the upper edge of the upper window, and sensor P6 was in the field of view in the lower window 10 mm below the horizontal bridge separating the lower window from the middle one. The red dashed lines mark the SW positions determined from the moments of the onset of bubble deformation between two successive video frames (at time intervals of 25 μs). During SW propagation in inert BW (upper row with frames (a) to (d)), the deformation and collapse of bubbles behind its front, as expected, did not lead to any glow. When SW propagated in active BW, bright flashes of light from explosions of individual bubbles were clearly visible. The average SW velocity in the tube section under consideration, obtained from the processing of video records of experiments with inert and reactive gas bubbles, was 320 and 400 m/s, respectively. These values were very close to the SW velocities determined from the shock front arrival time at sensors P5 and P6: 310 and 430 m/s, respectively (see Figure 6a). In Figure 7, during the exposure time of one frame Δ*t* = 25 µs, the SW front traveled a distance Δ*x* ≈10 mm. Individual frames show all the bubbles reacted over a time interval of Δ*t* and were located in a band with a width of Δ*x*. It is interesting that the luminous exploded bubbles were located at a distance of 10–20 mm behind the SW front; i.e., their self-ignition delay was 25–50 μs. Thus, the pressure fluctuations in the records of Figure 3 and Figure 4 were associated with shock-induced fluctuations of gas bubbles, and the peak pressure fluctuations in Figure 4 were associated with bubble explosions.

Figure 8 shows a sequence of video frames made at a speed of 500,000 fps in one of the experiments with active BW. Based on these frames, one can trace the dynamics of compression of an individual gas bubble behind the SW and determine the moment of its explosion and subsequent thermal expansion. Red circles mark two bubbles used to trace the entire process of bubble–SW interaction. Bubbles with an initial diameter *d*_0_ ≈ 2.5–3.5 mm (frames 1 and 15 in Figure 8) decrease in size to *d* ≈ 0.5–1.0 mm before explosion (frames 13 and 28), i.e., the bubble size decreases by a factor of *d*_0_/*d* ≈ 3.5–5 before explosion. After explosion, bubbles thermally expand to sizes exceeding the initial size: *d* ≈ (1.5–2)*d*_0_. The bubbles which explode in frames 14 and 15 are compressed in a propagating SW during 18–26 µs, which is in good agreement with the estimate made earlier in this paper based on the positions of the SW front and the glow. The time of intense chemical transformation (luminescence time) does not exceed the exposure time of a single frame (2 μs). The average rate of bubble collapse can be estimated from the change in the bubble size over the time interval from the moment of SW arrival at a bubble to bubble explosion, and it reaches 40–60 m/s. After explosion, the rate of bubble expansion reaches 100–125 m/s. It is worth noting that the error in calculating the sizes and rates is estimated at 30%–50% due to the limited spatial resolution. If one assumes that bubble compression is adiabatic, the gas temperature, *T*, inside bubbles can be estimated as *T* = *T*_0_(*d*_0_/*d*)^3(γ−1)^. At *d*_0_/*d* ≈ 3.5–5, the estimated temperature is higher than 1100 K. At such a temperature and a pressure of ≈7 MPa in the SW (see Figure 3 and Figure 4), the self-ignition delay of undiluted stoichiometric acetylene–oxygen mixture is several microseconds [47]. It should be noted, however, that judging by the registered glow, not all bubbles explode, which indicates the presence of heat losses and nonadiabatic bubble compression. One of the possible reasons for this effect is the instability of the bubble surface due to its uneven shock loading, which leads to the intensification of interfacial heat transfer. In order to register the development of instability, it was necessary to increase the spatial resolution of video filming and conduct experiments with weaker SWs.

As an example, Figure 9 shows video frames of the behavior of a single air bubble in inert BW with α = 2% when interacting with a relatively weak SW (amplitude Δ*P* = 0.13 MPa, propagation velocity *D* = 100 m/s, low-frequency speed of sound *C* = 85 m/s [48], Mach number M = 1.2). The passage of the SW through the bubble is seen to induce the formation of a cumulative jet that penetrates the bubble (frames 14 to 20). This phenomenon was discussed earlier in the literature, e.g., in [49]. Obviously, under such conditions, the intensity of interfacial heat transfer increases and the self-acceleration of chemical reactions leading to bubble explosion becomes more difficult.

Another possible reason for the observed effect is the nonuniform distribution of bubbles in water. Closely spaced bubbles influence each other even at a low volumetric gas content. As an example, Figure 10 shows video frames of deformation of two closely spaced bubbles in the same experiment as in Figure 8. The size of the bubble in the foreground is about 2.5 mm (the average size of the ellipsoid along two semiaxes), and there is a larger bubble with a diameter of about 4.5 mm in the background. In this case, the effect of bubble “piercing” by cumulative jets is much less pronounced, and after a single compression–expansion phase, the bubbles acquire the shape of a “sphere” with a strongly perturbed surface.

Figure 11 shows fragments of video frames for the same experiment as shown in Figure 7e–h relevant to active BW. As the SW propagated from top to bottom, several closely spaced bubbles were first compressed (frames 1–4). Then, three bubbles with a diameter of *d* ≈2 mm exploded almost simultaneously (frame 5). Thereafter, a larger bubble with a diameter of *d* ≈3.5 mm exploded with a time delay (frame 6), despite it being subjected to shock compression earlier, being somewhat upstream with respect to the two bubbles that exploded earlier. Furthermore, this bubble expanded (frame 7), shrank again (frame 8) and expanded again (frames 9–11), taking the shape of a parachute (frames 12–14), and then broke up into small fragments (frames 15–20).

The processing of video records made it possible to gain information on the shock-induced motion of individual bubbles behind the traveling SW. Thus, Figure 12a,b show the measured time histories of the bubble velocity in inert and active BW, respectively. In both cases, the bubble initial position was chosen near pressure sensor P6. The corresponding records of pressure sensor P6 are also shown in Figure 12a,b. The pressure records were averaged over time intervals equal to the exposure time of video frames (25 μs). The pressure profiles behind the SWs in both cases were almost identical. The initial ascending parts of curves *U*_b_(t) and their maxima (100–200 µs after SW arrival) corresponded to the velocity of the initial bubble, whereas the plateaus following the decline of the curves corresponded to the velocity of bubble microfragments formed after shock-induced bubble fragmentation. The plateau at the *U*_b_(*t*) curves represented the equilibrium velocity of phases (gas and water) behind the traveling SW. The bubble velocity fluctuations on the plots were the results of errors in determining bubble coordinates during data digitization. Quantitatively, in both cases, the velocities of initial bubbles in the SWs reached the value of ≈30 m/s, while the velocities of bubble microfragments in Figure 12a,b decreased to ≈17 m/s and ≈13 m/s, respectively, after about 1 ms after the passage of the SW. With additional averaging over time (in the interval of 0.25 ms after SW arrival) and over the ensemble of bubbles, the average bubble velocity in the tests of Figure 12a,b was 24 m/s and 25 m/s, respectively. Thus, in relatively weak SWs (at *P*_0,HPC_ = 0.6 MPa), the characteristic velocities of chemically inert and active bubbles in BW with α = 2% were approximately identical; i.e., bubble explosions did not contribute much to the SW-to-BW momentum transfer.

### 3.2. Shock Wave Package Propagation in Bubbly Water

In all experiments with SW package propagation in inert BW, a column ≈903 mm high with α ranging from 2% to 16% and air bubbles of mean diameter *d*_0_ = 3–4 mm were formed in the measuring section. The air and water were at room temperature. Figure 13 shows typical records of several signals: a signal triggering the video camera (a), the signal of electronic shutter (b), the record of pressure sensor P2 (c), and the record of pressure sensor P3 (d) installed in BW at a depth of 95 mm from the free surface of the BW column and at a distance of 50 mm from the exit sections of acceptor detonation tubes. The blue vertical line corresponds to the moment of time *t*_0_ when the countdown of video frames is launched. The error of determining this point in time depends on the sampling frequency (±1 μs at 1 MHz) and the trigger response time (±5 μs). The average detonation velocity in the acceptor detonation tubes was 2100 ± 150 m/s and the pressure behind the front of the detonation wave reached 4–6 MPa (estimated from the maximum pressure recorded by sensor P2 after applying the low-pass filter with a cutoff frequency of 100 kHz; see Figure 13).

The procedure of determining the detonation velocity in the acceptor detonation tubes was as follows. The pressure record (see signal (c) in Figure 13) was used to determine the moment of time when a detonation wave reached pressure sensor P2 (see Figure 1b). Then, using the signal of the electronic shutter (signal (b) in Figure 13), the number of the video frames corresponding to this moment in time was determined. Based on the known number of video frames (37 pcs) from the initial moment of time and the time interval between frames (20 μs at a shooting rate of 50,000 fps), the time *t*_1_ when the detonation wave arrived at sensor P2 (green vertical line in Figure 13) was determined as *t*_1_ = 37 × 20 = 740 μs. The time moment when the detonation wave exited from the tube (*t*_2_) was determined from the video record (see below).

For the experiment presented in Figure 13, *t*_2_ = 68 × 20 = 1360 µs. Knowing the distance between pressure sensor P2 and the tube cut, *L* = 1265 mm, it is possible to determine the average detonation propagation velocity *D* = *L*/(*t*_2_ − *t*_1_), which in this case was equal to 2040 ± 60 m/s. The time interval between SW pulses can be estimated from the distance (≈300 mm) between adjacent acceptor detonation tubes and from the average detonation velocity (≈2100 m/s). Calculation gives a value approximately equal to 140 µs (frequency ≈ 7 kHz). In the experiment of Figure 13, the same time interval separates the pressure maxima at sensor P3 (signal (d) in Figure 13). Note that the fourth pressure maximum in the record of sensor P3 arises due to SW propagation not only in the longitudinal but also in the transverse direction in the BW column.

Table 1 shows the measured values of the velocity of the leading front of the SW package in BW with different α at three measuring segments: between pressure sensors P3 and P4, P4 and P5, and P5 and P6. The front velocity *D*ij was calculated as the known distance *L*_ij_ between the pressure sensors (see Figure 1b) divided by the time interval required for the front to travel between the sensors, Δ*t*_ij_: *D*_ij_ = *L*_ij_/Δ*t*_ij_. The maximum error in measuring the front velocity is estimated at 3%. It follows from Table 1 that the velocity of the leading front of the SW package exceeds the corresponding low-frequency speed of sound in BW by a factor of 1.5–2; i.e., the front propagation velocity is supersonic. In addition, it is seen that the velocity of the leading front of the wave package at the position of sensor P3 decreases with α.

Figure 14 shows examples of records of pressure sensor P3 for different values of α. The following specific features of SW package propagation in BW can be highlighted. Firstly, the shape of the leading front of the wave package in BW differs significantly from the shape of an SW front in gas: the time of pressure rise in the front of the SW package in BW varies from 0.2 (Figure 14a–c) to 0.4 ms (Figure 14d). Secondly, as α increases, the time intervals between the individual pressure maxima in the successive SWs decrease, i.e., the SWs catch up with each other and merge, and the profile of the SW package is smoothed and approaches a triangular shape. These effects are associated with a stepwise decrease in the volumetric gas content after each successive SW and, accordingly, with an increase in the low-frequency speed of sound in BW ahead of each successive SW (see Table 1). Thirdly, with an increase in α, the SW amplitudes in the SW package decrease, which is consistent with the decrease in the propagation velocity of its leading front noted above (see Table 1).

High-speed video filming makes it possible to measure the penetration depth of gaseous detonation products, *L*_p_, and the velocity of BW behind the SW package, *u*_cs_, by tracking the motion of the contact surface “detonation products–BW” [35]. Table 2 shows the measured depths of product gas jets and the contact surface velocities at different distances (20 mm (*u*_sc,20-mm_), 40 mm (*u*_sc,40-mm_), and 60 mm (*u*_sc,60-mm_)) from the cutoff of acceptor detonation tubes, which was determined from video frames. It can be seen that with an increase in α from 2% to 16%, the longitudinal velocity of the contact surface near the cut of the acceptor detonation tubes (20 mm) decreases. However, at greater depths (40 and 60 mm), the longitudinal velocity of the contact surface increases by a factor of 2 with such an increase in α. Interestingly, as α increases, the depth of penetration of product gas jets changes insignificantly.

For illustration, Figure 15 shows video frames of the penetration of detonation products from three successive detonation waves into BW at α = 2%. The first detonation wave enters BW from the central acceptor tube (frame #70); then, with a delay of ≈140 µs (frame #77), the second detonation wave leaves the right acceptor tube, and after another ≈140 µs (frame #84), the third detonation wave leaves the left acceptor tube. With the growth of the product gas bubble, the height of the BW column noticeably increases (in frame #200, it comes close to the upper border of the image). It is interesting that the shape of the product gas bubble does not show any signs of asymmetry: over time, the product gas bubble is elongated mainly in the longitudinal direction, while its transverse dimensions change insignificantly.

The results of experiments show that the use of high-frequency SW pulses in the PDH is pointless because the interference of pulses impairs the SW-to-BW momentum transfer. For efficient operation of the PDH, it is necessary to maintain the required SW intensity and the required level of volumetric gas content (≈20–25% [35]) in its water guide. The SW intensity is mainly determined by the fuel mixture used. Thus, the detonation velocity in propane–oxygen and propane–air mixtures is ≈2360 and ≈1800 m/s, respectively, and the overpressure at the Chapman–Jouguet point is ≈3.52 and ≈1.73 MPa, respectively. As for the volumetric gas content, its required level can be provided either by the forced aeration of seawater or by using gaseous detonation products of the previous cycle. In the former case, for implementing the operation process, the frequency of SW pulses must be such as to exclude cycle interference; i.e., by the beginning of the next cycle, both the SW and the product gas bubble from the previous cycle must leave the water guide, whereas the water guide must be filled with a new charge of BW. With a small-scale water guide length of 0.5 m and a BW velocity in the water guide of 25 m/s [39] (at an approaching water velocity of 10 m/s), the required operation frequency should not exceed ≈50 Hz. In the latter case, the required PDH operation frequency must exceed ≈50 Hz, but a new cycle must begin after the water guide is filled (more or less uniformly) with the detonation products of the previous cycle. If one takes into account that the volume occupied by detonation products in the water guide should be at a level of ≈20–25%, then the operation frequency should not exceed ≈60 Hz.

## 4. Conclusions

Compressible bubbly water can be used for generating a propulsive force in large, medium, and small-scale underwater thrusters referred to as pulsed detonation hydroramjets. Thrust in such thrusters is produced due to the acceleration of bubbly water in a flow-through water guide by periodic shock waves and product gas jets generated by pulsed detonations of a fuel–oxidizer mixture in a detonation tube inserted into the water guide. The experimental studies reported herein were aimed at exploring two possible directions of the improvement of thruster performances, namely, (1) the replacement of chemically nonreacting gas bubbles by chemically reactive bubbles, and (2) the increase in the pulsed detonation frequency from tens of hertz to some kilohertz.

Experiments on single shock wave propagation in bubbly water with chemically reactive gas bubbles revealed the possibility of obtaining a bubbly quasidetonation wave propagating at the velocity, exceeding by up to 50% the velocity of shock wave propagation in bubbly water with chemically nonreacting gas due to shock-induced energy release caused by bubble explosions. Such quasidetonation waves were obtained in bubbly water with bubbles of acetylene–oxygen mixture 1.5–4 mm in diameter without thickening additives, at volumetric gas content up to 10% and at distances less than 1 m. The registered increase in the shock wave velocity can be used for increasing the thruster operation frequency and thrust.

Experiments on high-frequency (≈7 kHz) shock wave package propagation in bubbly water with air bubbles of different diameters (3–4 mm) at volumetric gas content up to 16% showed that the use of high-frequency shock wave pulses is pointless for the thrusters under consideration: the resulting interference of pulses worsens the shock wave-to-bubbly water momentum transfer. On the one hand, the shock waves penetrating bubbly water quickly merge, feeding each other and increasing the velocity of bubbly water, while on the other hand, the initial gas content of bubbly water for each subsequent shock wave decreases and, accordingly, the efficiency of shock wave-to-bubbly water momentum transfer decreases. Estimates show that for a small-scale water guide of 0.5 m length, the optimal frequency of detonation pulses is about 50–60 Hz.

## Figures and Tables

**Figure 1 micromachines-13-01553-f001:**
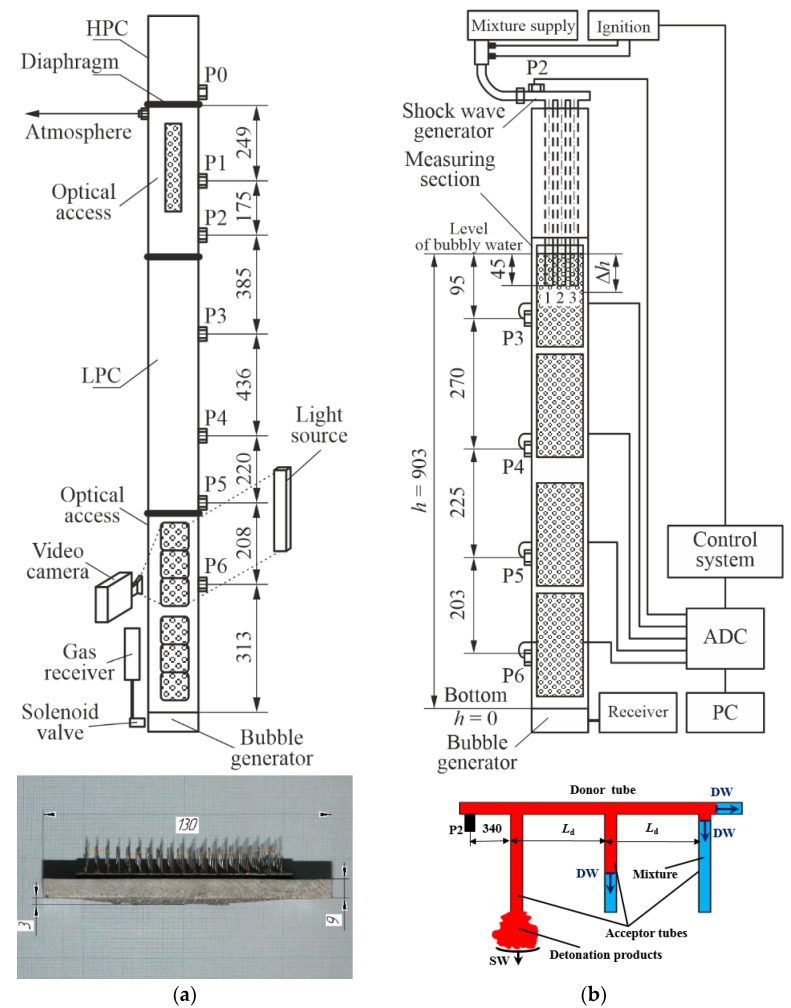
Schematics of test rigs for studies of (**a**) single shock wave and (**b**) shock wave package propagation in bubbly water. Inserts show the photo of bubble generator and schematic of shock wave generator. Dimensions are given in millimeters.

**Figure 2 micromachines-13-01553-f002:**
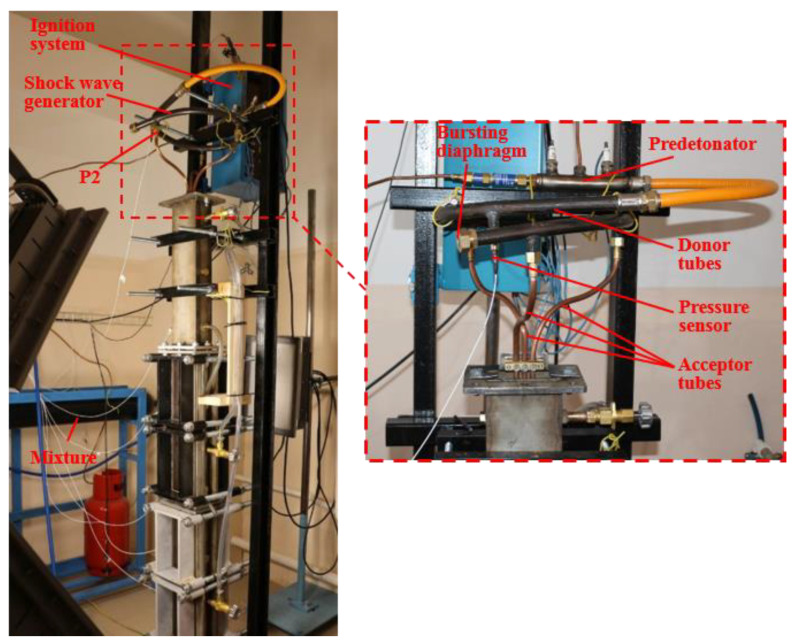
Photograph of the test rig for studies of shock wave package propagation in bubbly water with the explosive view of the SW generator (insert).

**Figure 3 micromachines-13-01553-f003:**
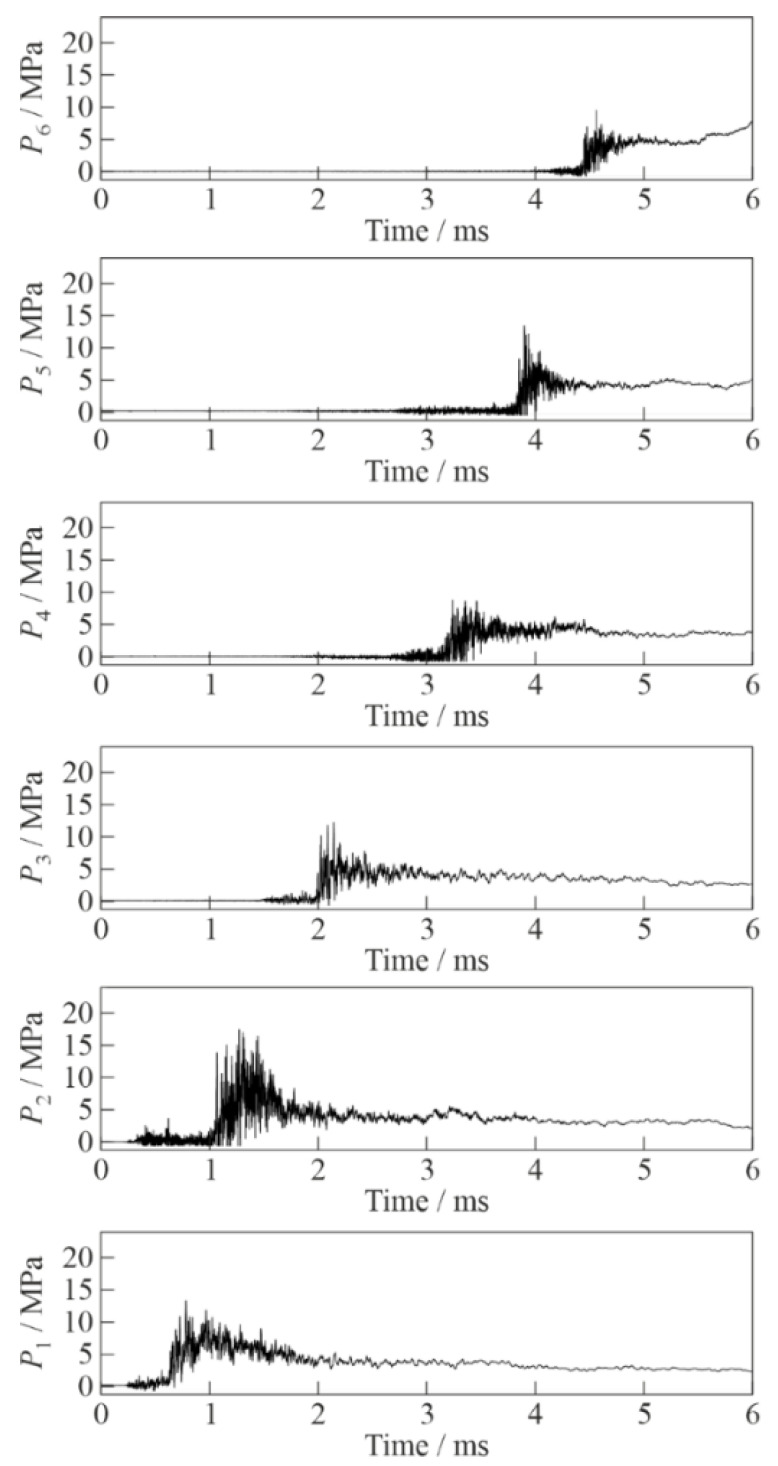
Records of pressure sensors P1 (bottom) to P6 (top) in the experiment with inert bubbly water (α = 2%, *P*_0,HPC_ = 0.6 MPa).

**Figure 4 micromachines-13-01553-f004:**
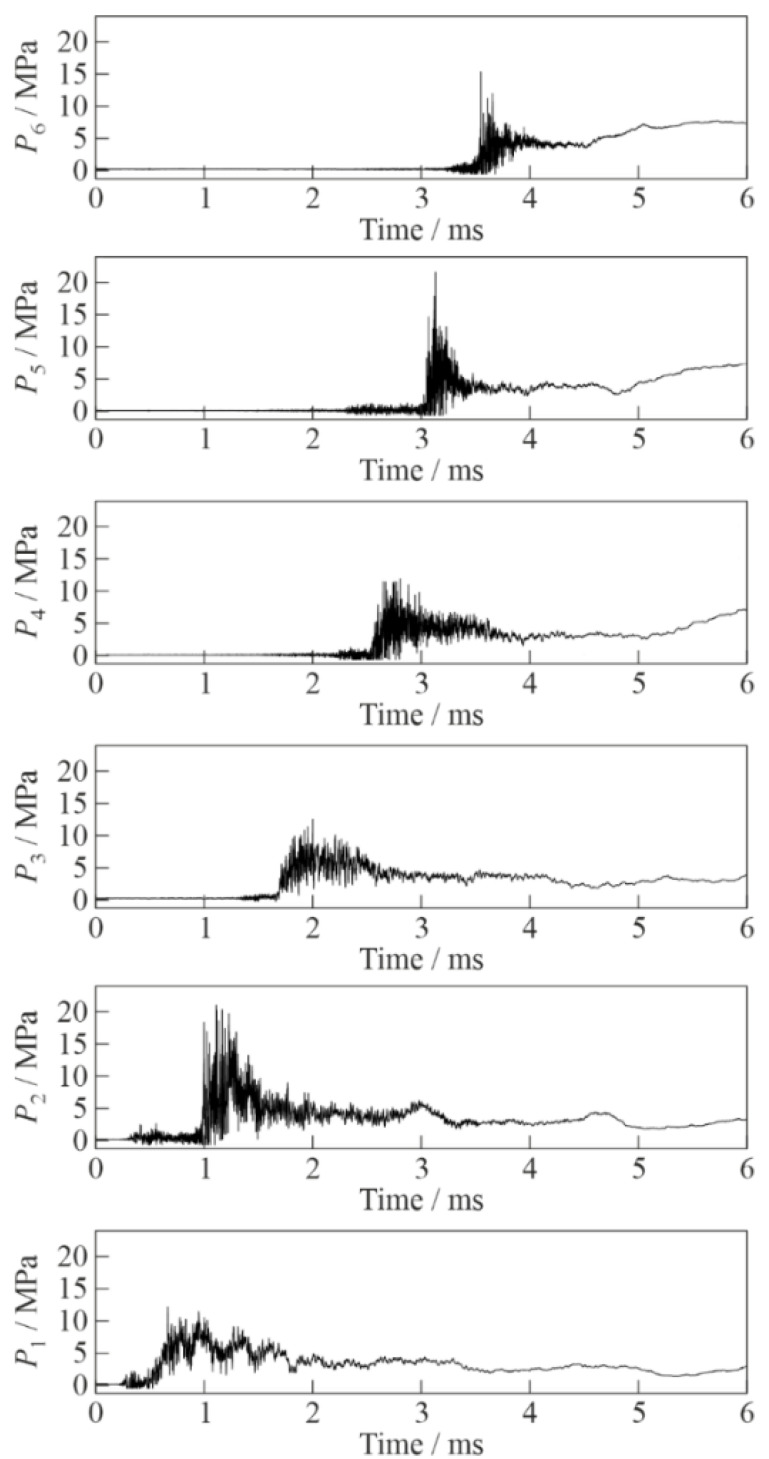
Records of pressure sensors P1 (bottom) to P6 (top) in the experiment with active bubbly water (α = 2%, *P*_0,HPC_ = 0.6 MPa).

**Figure 5 micromachines-13-01553-f005:**
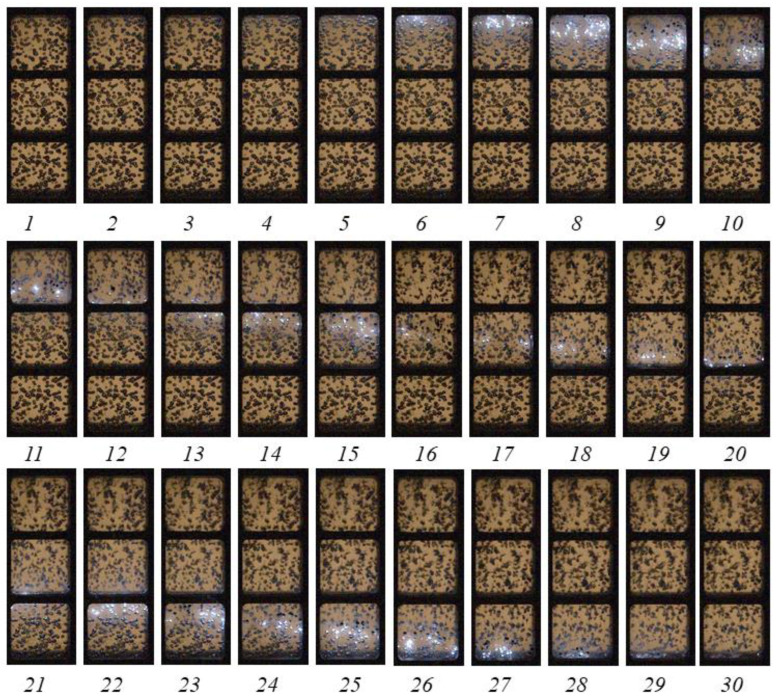
Video frames of propagation of bubbly quasidetonation in active bubbly water at α = 2% and *P*_0,HPC_ = 0.6 MPa. Frames 1 to 30 are numbered sequentially from the moment of ignition. Frame size 160 × 448 pixels, video recording rate 40,000 fps.

**Figure 6 micromachines-13-01553-f006:**
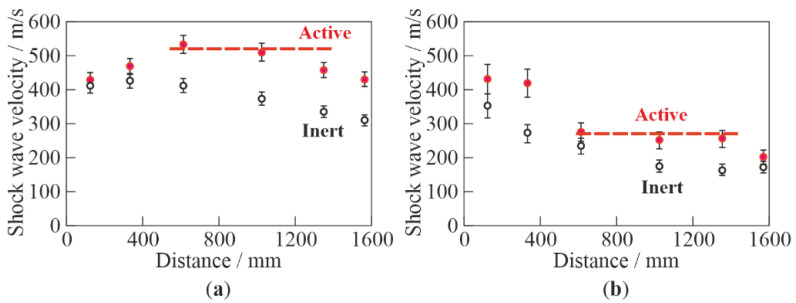
The SW velocity vs. distance (*P*_0,HPC_ = 0.6 MPa) in inert and active bubbly water at (**a**) α = 2% and (**b**) 10%.

**Figure 7 micromachines-13-01553-f007:**
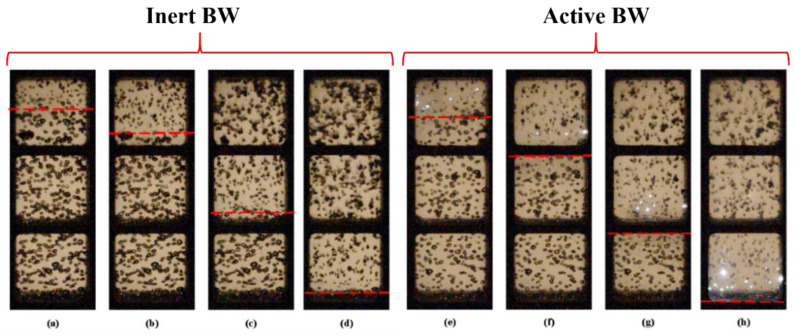
Video frames of SW propagation in inert (**a**–**d**) and active (**e**–**h**) bubbly water at α = 2% and *P*_0,HPC_ = 0.6 MPa. Time countdown is from the moment of ignition. Frame size 160 × 448 pixels, video recording rate 40,000 fps. The red dashed lines show the positions of the SW front at (**a**) *t* = 4.091 ms; (**b**) 4.166 ms; (**c**) 4.366 ms; (**d**) 4.591 ms; (**e**) 3.250 ms; (**f**) 3.325 ms; (**g**) 3.475 ms; and (**h**) 3.650 ms.

**Figure 8 micromachines-13-01553-f008:**
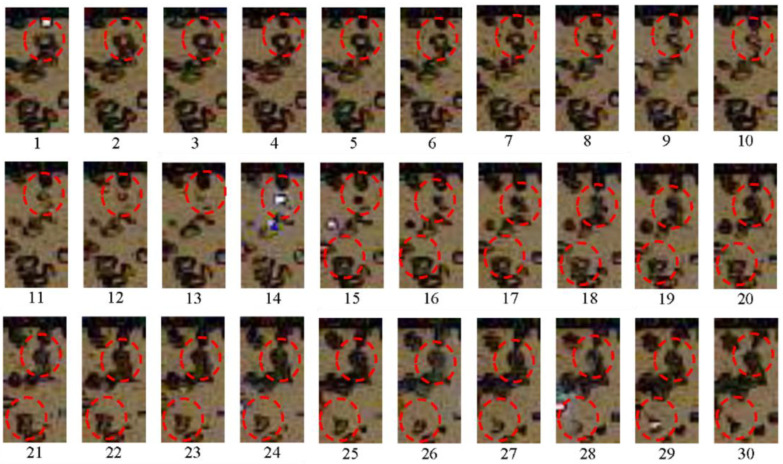
Video frames of SW propagation in active bubbly water at α = 3% and *P*_0,HPC_ = 0.55 MPa. The first frame corresponds to *t*_1_ = 4.9 ms, the following frames (numbered from 1 to 30 are made with an interval of Δ*t* = 2 µs. The frame size is 24 × 48 pixels. The top edge of the images is 6.5 cm below pressure sensor P5.

**Figure 9 micromachines-13-01553-f009:**
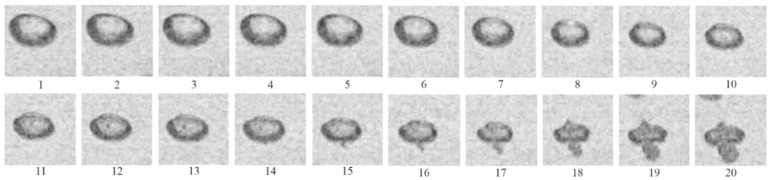
Video frames of SW propagation in inert bubbly water at α = 2%, Δ*P* = 0.13 MPa and *D* = 100 m/s. Bubble size *d*_0_ = 2.8 mm. Frames are numbered sequentially from 1 to 20 with an interval of Δ*t* = 50 µs.

**Figure 10 micromachines-13-01553-f010:**
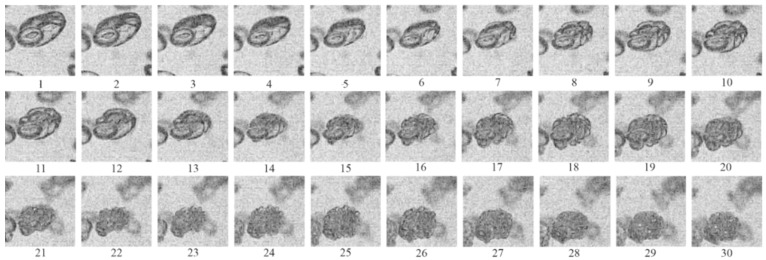
Video frames of SW propagation in inert bubbly water at α = 2%, Δ*P* = 0.13 MPa, and *D* = 100 m/s. The sizes of the bubbles in the foreground and background are *d*_0_ = 2.5 mm and 4.5 mm, respectively. Frames are numbered sequentially from 1 to 30 with an interval of Δ*t* = 50 µs.

**Figure 11 micromachines-13-01553-f011:**
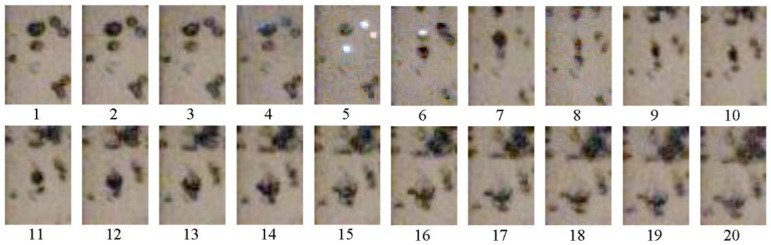
Fragments of video frames of SW propagation in active bubbly water at α = 2% and *P*_0,HPC_ = 0.6 MPa. The first frame corresponds to *t*_1_ = 3.175 ms. Frames are numbered sequentially from 1 to 20 with an interval of Δ*t* = 25 µs. Time countdown starts from the moment of ignition in the HPC.

**Figure 12 micromachines-13-01553-f012:**
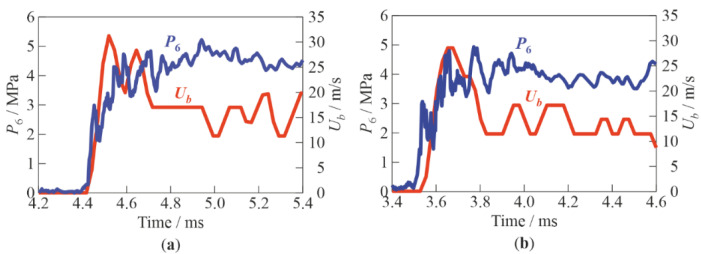
Measured time histories of pressure *P*_6_ registered by pressure sensor P6 and bubble velocity *U*_b_ near this sensor at α = 2% and *P*_0,HPC_ = 0.6 MPa: (**a**) inert; (**b**) active bubbly water.

**Figure 13 micromachines-13-01553-f013:**
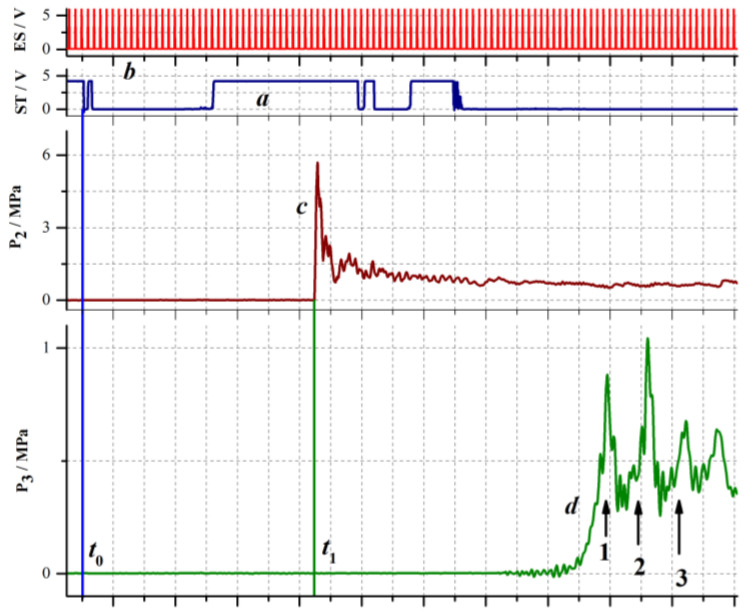
Typical recorded signals in an experiment with SW package propagation in BW at α = 2%: (a) a signal triggering the video camera, ST; (b) electronic shutter, ES; (c) pressure sensor P2, and (d) pressure sensor P3; numbers 1, 2, and 3 indicate the successive SWs.

**Figure 14 micromachines-13-01553-f014:**
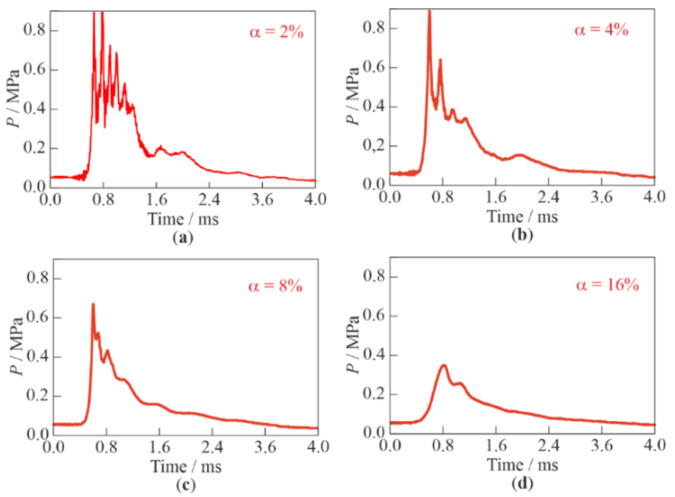
Examples of records of pressure sensor P3 during the propagation of an SW package of three SWs in bubbly water with different α: (**a**) 2%; (**b**) 4%; (**c**) 8%; and (**d**) 16%.

**Figure 15 micromachines-13-01553-f015:**
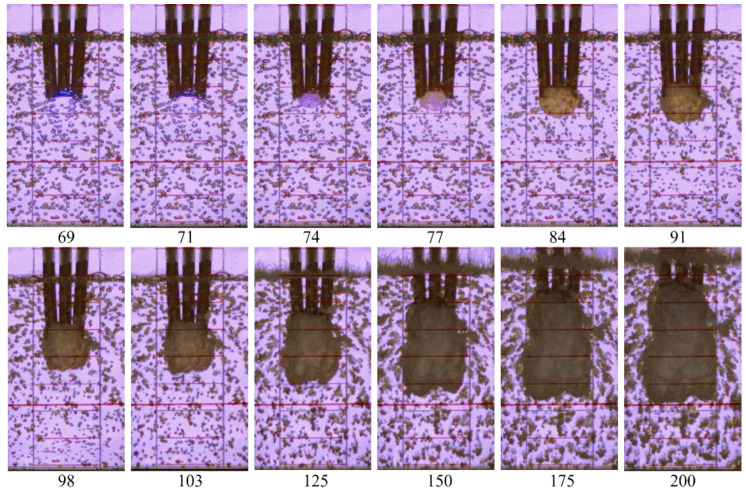
Video frames of the outflow of pulsed jets from three acceptor detonation tubes to bubbly water with α = 2%; frame size 168 × 320 pixels, frame rate 50,000 fps; shutter speed 18 µs.

**Table 1 micromachines-13-01553-t001:** Measured values of the SW package front velocity, *D*_ij_, at measuring segments ij in bubbly water with different α.

α, %	*C*, m/s	*D*_34_, m/s	*D*_45_, m/s	*D*_56_, m/s
2	85	180	140	126
4	61	115	92	82
8	44	75	60	55
16	32	50	40	40

**Table 2 micromachines-13-01553-t002:** Measured depths of product gas jet penetration, *L*_p_, and velocities of the contact surface “detonation products–bubbly water” for SW packages in bubbly water with different α.

α, %	*L*_p_, mm	*u*_sc,20-mm_, m/s	*u*_sc,40-mm_, m/s	*u*_sc,60-mm_, m/s
2	66	35	13	3
4	56	39	13	4
8	56	35	20	4
16	57	26	26	6

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
