# Peer review of "Interaction of Shock Waves with Water Saturated by Nonreacting or Reacting Gas Bubbles"

_micromachines, 2022, doi:10.3390/mi13091553_

Round 1

Reviewer 1 Report

This manuscript requires improvement before it can be considered for publication.

--Improve abstract content and organization (i.e. first line in abstract needs improvement, etc)

--Improve English grammar and manuscript organization (i.e. Word choice, sentence structure, plural (tens, etc), avoid the use of "we", "our", etc; avoid 1 sentence paragraphs; improve content; use of proper terms,;  avoid starting consecutive sentences with the same word, proper spacing, etc)

--avoid lumping references...discuss each individually

--improve experiment and component  details

--captions should have additional details

--figure 2 should also include schematic that shows the components of the experiment

--all figures and tables need a detailed discussion within the text

-- fig 2, 3, 4, 7,8,9 13 etc show multiple items ...each should be identified (a), (b), (c) etc

--are all figures necessary?

--some figures  fig 7, 8, 9 ,etc have multiple figures that look the same ...please discuss

--perhaps the various figures should have a time associated with it rather than a number

--improve the conclusion  discussion 

--are all variables defined?

--provide an analytical description of the problem 

--provide more current (less than 5 yrs old) references...currently there are only 7 of the 40 references that are current

--perform a sensitivity and error analysis

--what is the accuracy of the various components

--detail the novel aspects of the manuscript

--is the data in fig 12 digital or graphic?

--

Author Response

We are grateful to the reviewer for valuable comments. We made our best to follow all the comments. All changes in the revised manuscript are marked in yellow.

This manuscript requires improvement before it can be considered for publication.

--Improve abstract content and organization (i.e. first line in abstract needs improvement, etc)

We have shortened the first sentence of the abstract and reformulated the abstract using shorter sentences.

--Improve English grammar and manuscript organization (i.e. Word choice, sentence structure, plural (tens, etc), avoid the use of "we", "our", etc; avoid 1 sentence paragraphs; improve content; use of proper terms,;  avoid starting consecutive sentences with the same word, proper spacing, etc)

We have asked our native English-speaking colleague to check the manuscript in terms of grammar and made our best to follow the reviewer’s suggestions.

--avoid lumping references...discuss each individually

We replaced lumping references by brief description of the contribution of each individual reference.

--improve experiment and component details

To address this comment, we have added the photograph of bubble generator to Figure 1a and added some more details on the test rig of Figure 1b.

--captions should have additional details

We have extended figure captions where applicable.

--figure 2 should also include schematic that shows the components of the experiment

We have included such a schematic to Figure 1b.

--all figures and tables need a detailed discussion within the text

We have extended the discussion of some figures and slightly rearranged the manuscript by adding a new Figure 5 illustrating the bubbly quasidetonation.

-- fig 2, 3, 4, 7,8,9 13 etc show multiple items ...each should be identified (a), (b), (c) etc

We have added the word “insert” to the caption of Figure 2. As for other figures mentioned by the reviewer, digits denote the frame number, which is convenient for calculating time intervals.

--are all figures necessary?

We think all video frames are necessary for the sake of clarity.

--some figures  fig 7, 8, 9 ,etc have multiple figures that look the same ...please discuss

Actually, these figures illustrate different phenomena during shock wave – bubbly water interaction, namely, bubble compression and ignition (former Figure 7), instability of bubble surface and bubble “piercing” by a cumulative jet (former Figure 8), and collective effects on bubble instability (former Figure 9).

--perhaps the various figures should have a time associated with it rather than a number

It is more convenient to refer to a frame number rather than to frame time. When we show frame numbers, we specify the value of time interval between the frames. Otherwise, if the time interval is nonuniform, we indicate time in seconds, like in Figure 7.

--improve the conclusion  discussion 

We have reformulated some sentences in the Conclusions.

--are all variables defined?

We have checked that all variables are defined and listed in the nomenclature.

--provide an analytical description of the problem 

 The first paragraph in the Introduction claims that “Theoretically, the PDH thrust is proportional to the operation frequency, which depends on both the SW velocity in BW and on the frequency of pulsed detonations.” Then we write (Line 114) “The possible directions of the improvement of PDH thrust performances are to use BW with bubbles of chemically active rather than inert gas (i.e., to increase the SW velocity in BW) and to increase the frequency of pulsed detonations from tens of hertz to some kilohertz.” Finally, the last paragraph in the Introduction reads: “In view of these two possible directions of the improvement of PDH thrust performances, the objective of the present work was twofold. On the one hand, the phenomenology of single SW propagation in pure water saturated with nonreacting or reacting bubbles, specific features of SW-to-BW momentum transfer, and the phenomenology of propagating bubbly detonations was studied experimentally. On the other hand, the specific features of the interaction of high-frequency (~7 kHz) SW pulses with water saturated with air bubbles was studied experimentally. These objectives, as well as the obtained results, are the novel and distinctive features of the present work.” We believe, this is sufficient for problem statement.

--provide more current (less than 5 yrs old) references...currently there are only 7 of the 40 references that are current

We have added 7 more recent publications in the list of references.

--perform a sensitivity and error analysis

We have additionally specified the error of determining the gas volume fraction and pressure.

--what is the accuracy of the various components

When applicable, we always mention the estimated error in the manuscript, see, e.g., section 3.2.

--detail the novel aspects of the manuscript

The last paragraph of the Introduction claims the novel aspects of the manuscript: “These objectives, as well as the obtained results, are the novel and distinctive features of the present work.”

--is the data in fig 12 digital or graphic?

It is digital. We have replotted this figure (now Figure 13) in a standard format.

Reviewer 2 Report

The paper Interaction of shock waves with water saturated by nonreacting or reacting gas bubbles is within the scope of micromachines. Following recommendations should be considered before publication :

1 - Adding more papers from micromachines in literature.

2- Instead of group citing please decribe one by one. 14–22 is not acceptable.

3 - Effects of propagating bubbly detonations is not discussed physically. 

4- The accuracy of controlling device should be discussed.

5- Reproduce Figure 12 in the normal format.

6- Please describe more about : "The time interval between SW pulses can be estimated from the distance (~300 mm) between adjacent acceptor detonation tubes and from the average detonation velocity (~2100 m/s)".

7- Peaks on Figure 13 should be discussed.

8- The frequency effects on bubble stability is not well disclosed.

9 - The reason of the changing in efficiency of shock-to-water momentum transfer should be discussed.

10- Detonation waves on the surface of the bubble should be illustrated in separate figure.

Author Response

We are grateful to the reviewer for valuable comments. We made our best to follow all the comments. All changes in the revised manuscript are marked in yellow.

The paper Interaction of shock waves with water saturated by nonreacting or reacting gas bubbles is within the scope of micromachines. Following recommendations should be considered before publication:

1 - Adding more papers from micromachines in literature.

We have added a recent review paper [2] to the list of references. Also, we have added reference [3] to show more potential applications.

2- Instead of group citing please decribe one by one. 14–22 is not acceptable.

Done.

3 - Effects of propagating bubbly detonations is not discussed physically. 

We have added a new Figure 5 and some discussion to illustrate the propagation of bubbly detonation.

4- The accuracy of controlling device should be discussed.

In section 3.2, the response time of the control system was mentioned in the original manuscript.

5- Reproduce Figure 12 in the normal format.

Done. Now it is Figure 13.

6- Please describe more about : "The time interval between SW pulses can be estimated from the distance (~300 mm) between adjacent acceptor detonation tubes and from the average detonation velocity (~2100 m/s)".

To clarify it, we have added an insert to Figure 1b.

7- Peaks on Figure 13 should be discussed.

Peaks in this figure (now it is Figure 14) have the same meaning as in the present Figure 13 and correspond to the successive shock waves. Among the experiments with a gas volume fraction of 2% we have found a better case with more pronounced peaks and replaced Figure 14a for the sake of clarity.

8- The frequency effects on bubble stability is not well disclosed.

In this work, we did not study the effect of shock wave frequency in the package on bubble stability.

9 - The reason of the changing in efficiency of shock-to-water momentum transfer should be discussed.

The effect of gas volume fraction in bubbly water on shock-to-water momentum transfer was studied in detail in our previous publication in Int. J. Multiphase Flow [35]. In the present manuscript, we do not address this issue.

10- Detonation waves on the surface of the bubble should be illustrated in separate figure.

 Unfortunately, due to the limited space and time resolution of video recording, we could not visualize the details of strong shock (detonation) interaction with an individual bubble. We could do it only with weak shocks (see Figure 9), and it is mentioned in the manuscript (see a sentence in Section 3.1 marked in green)

Round 2

Reviewer 1 Report

The improved version of the manuscript has addressed most of my concern